# Treatment of Lumbar Degenerative Disease with a Novel Interlaminar Screw Elastic Spacer Technique: A Finite Element Analysis

**DOI:** 10.3390/bioengineering10101204

**Published:** 2023-10-16

**Authors:** Zebin Huang, Shu Liu, Maodan Nie, Jiabin Yuan, Xumiao Lin, Xuerong Chu, Zhicai Shi

**Affiliations:** 1Department of Orthopedics, Changhai Hospital, Naval Medical University, Shanghai 200433, China; zbhuang0105@163.com (Z.H.); acoliushu@163.com (S.L.); smmuyuan@163.com (J.Y.); superlin0426@163.com (X.L.); 2School of Biomedical Engineering, Shanghai Jiao Tong University, Shanghai 200240, China; michelle0702@sjtu.edu.cn; 3Daxing Hospital, Xi’an 710082, China; xuerongchu@126.com

**Keywords:** lumbar degenerative disease, translaminar transfacet technique, non-fusion fixation, elastic rod, finite element analysis

## Abstract

A novel interlaminar elastic screw spacer technique was designed to maintain lumbar mobility in treating lumbar degenerative diseases. A validated finite element model of L4/5 was used to establish an ISES-1/2 model and an ISES-1/3 model based on different insertion points, a unilateral fixation model and a bilateral fixed model based on different fixation methods, and a Coflex-F model based on different implants. The elastic rods were used to fix screws. Under the same mechanical conditions, we compared the biomechanical characteristics to investigate the optimal entry point for ISES technology, demonstrate the effectiveness of unilateral fixation, and validate the feasibility of the ISES technique. Compared to ISES-1/3, the ISES-1/2 model had lower intradiscal pressure, facet cartilage stress, and posterior structural stress. Compared to the ISES-BF model, the ISES-UF model had lower intervertebral pressure, larger mobility, and smaller stress on the posterior structures. The ISES model had a similar intervertebral pressure and limitation of extension as the Coflex-F model. The ISES model retained greater mobility and reduced the stress on the facet cartilage and posterior structure compared with the Coflex-F model. Our study suggests that the ISES technique is a promising treatment of lumbar degenerative diseases, especially those with osteoporosis.

## 1. Introduction

Lower back pain is the leading cause of disability, resulting in significant healthcare burdens and costs each year. The prevalence of lower back pain varies from 1.4% to 20.0%, and the incidence ranges from 0.02% to 7.0% [1]. It is associated with age, occupation, education level, and lifestyle [2]. Lumbar degenerative diseases are the most common cause of lower back pain. The clinical manifestations primarily include sciatica and neurogenic intermittent claudication, which significantly impact the patient’s work ability and quality of life. Interspinous spacers (ISPs) have gained popularity as an alternative treatment for lumbar degenerative diseases, serving to decrease the intradiscal pressure (IDP) and maintain the intervertebral height [3]. Biomechanical studies have demonstrated that ISPs reduce hyperextension at the instrumented level, increase the dimensions of the neural foramen and spinal canal, and decrease the IDP [4]. However, the long-term clinical results of ISPs have not been satisfactory. This is primarily attributed to the weak structural strength of the spinous process and the unstable fixation provided by ISPs. Complications such as a spinous process fracture, ISPs loosening, and the breakage of the fixation device often necessitate revision surgery [4,5,6]. Hence, there is a need for a new device that can effectively slow down the progression of lumbar degenerative diseases while minimizing complications, particularly in patients with osteoporosis.

Trautwein et al. found that the vertebral lamina exhibits high strength, with a minimum disruption strength 2–5 times greater than that of the spinous process [7]. The translaminar transfacet (TT) technique capitalizes on this property to achieve robust screw-to-bone fixation [8]. In contrast to conventional rigid stabilization devices, the TT technique, which involves screw placement from the vertebral lamina to the contralateral facet joints, offers a less rigid spinal stabilization system that facilitates an increased load transfer across the spine [9]. However, it should be noted that the TT technique requires the fusion of the facet joint, which compromises its integrity and may lead to complications such as osteoarthritis, pseudarthrosis, and accelerated adjacent segment degeneration [10,11]. The successful application of the TT technique has demonstrated the feasibility of implanting screws into the vertebral lamina and has shown the stability of fixation using this approach. This achievement provides us with innovative inspiration to explore techniques for optimizing the treatment of lumbar spine degeneration. By utilizing the strength of the lamina to fix the screws, without causing damage to the facet joints, and by incorporating a specially designed elastic rod to support the adjacent screws, the interlaminar screw elastic spacer (ISES) technique aims to maintain intervertebral heights, enhance spinal mobility, and limit lumbar over-extension.

The screw insertion path of TT was used as a reference for the ISES technique, in which an interlaminar screw was inserted at the base of the spinous process, through the opposite lamina and ending near the facet joint, without disrupting the facet joint and preserving the mobility of the facet joint. The ISES technique consists of two interlaminar screws connected by an elastic rod, as shown in Figure 1. Both landmarks, a cephalad one-third (ISES-1/3) and one-half (ISES-1/2) of the spinolaminar junction, can be used as the insertion points for interlaminar screws, which have been shown to be safe [12], but a more optimal entry point is still controversial [12,13,14]. As a highly screw-dependent technology, it is necessary to investigate the optimal entry point.

Moreover, non-fusion dynamic fixation techniques can be utilized to allow varying degrees of lumbar motion by adjusting the elastic capacity of the connecting rods. Our team has designed a novel type of elastic rod with a unique dynamic structure, comprising a titanium hollow shell and a nickel–titanium ‘cable’ structure. We hypothesized that the elastic rod would maintain a certain level of spinal mobility while limiting excessive extension and reducing complications commonly associated with rigid fixation.

The ISES technique is a novel non-fusion technique that requires the experimental verification of its feasibility. Finite element (FE) simulation analysis is a numerical analysis method based on mathematical and physical principles. It allows for the systematic manipulation of individual input factor, such as force or moment, to assess its impact on various screw insertion techniques [15]. It is a widely used approach in the development of medical devices, enabling convenient design modifications [16]. In this study, an FE model was employed to investigate the optimal entry point for interlaminar screws, assess the feasibility of unilateral fixation, and identify potential complications associated with the ISES technique. After planning the content of the ISES technique, the mechanical feasibility of the ISES technique was verified by comparing it with the Coflex-F technique, which has similar principles and the same function. The main variables in this study include the range of motion (ROM) of the spinal segments, the IDP, the average von Mises stress at the facet cartilage, and the maximum von Mises stress of the posterior structure. The purpose of this study is to assess the feasibility of using ISES technology in the treatment of lumbar degenerative diseases and to discover potential complications.

## 2. Materials and Methods

### 2.1. Development and Validation of Intact Lumbar FE Model

The intact lumbar spine (INT) model used in this study is the same as the one used in our previous research [17]. The collagen fibers were represented using tension-only truss elements (T3D2) embedded in the intervertebral disc’s ground substance. The fibers had an average angle of ±30° to the endplates and consisted of eight layers [17]. The volume ratio of the annulus fibrosus and nucleus pulposus was 3:7 [18]. The cortical bone was 1 mm thick and the facet joint’s cartilage layer thickness was assumed to be 0.2 mm. Ligaments, including the anterior longitudinal ligament (ALL), posterior longitudinal ligament (PLL), interspinous ligament (ISL), supraspinous ligament (SSL), ligamentum flavum (LF), ligament intertransversarii (ITL), and capsular ligament (CL), were modeled as tension-only truss elements (T3D2). Node sharing was set between each ligament and its attachment point to the bone as well as between each vertebra and disc to increase the efficiency of modeling. The specific process of constructing the model has already been demonstrated in our previous research [17]. The element size was reduced until there was a change of less than 2% in the disc stress. The resulting model consisted of 604,487 elements of 1 mm in size. Detailed properties of all components in the lumbar spine model are outlined in Table 1.

To validate the FE model, we first applied pure compressive forces (150 N, 400 N, and 1000 N) separately to the model and obtained its IDP, which were then compared with in vitro experimental data in the published papers [16,19]. Next, 300 N and three different moments (3.0 Nm, 7.5 Nm, and 10 Nm) were applied to the model in different directions to simulate flexion, extension, lateral bending, and axial rotation. The ROMs were documented and compared to previous study results [20,21,22]. The more detailed validation process and results of the model have already been documented in our previous research [17].

### 2.2. Development of Surgical Lumbar FE Model

This study represents an initial exploration of the feasibility of the ISES technique. Figure 1 illustrates five three-dimensional lumbar models used in this study: the INT model, the ISES-1/2 model, the ISES-1/3 model, ISES-BF model, and the Coflex-F model. The validated INT model was adapted by incorporating the insertion of interlaminar screws or the Coflex-F device. In this study, the interlaminar screws were simplified as cylinders with a diameter of 4.5 mm, and the rod length was set to protrude 5 mm from the screw head [17]. The material of the internal elastic structure was nickel–titanium alloy, and the Young’s modulus and Poisson’s ratio of the model were set to 47 GPa and 0.3. The elastic rod was confined to the screw head and limited to all degrees of freedom, which simulates a tail nut locking the rod tightly to the screw head. The screw–bone interface was modeled using surface-to-surface contact, assuming a coefficient of friction of 0.8 to simulate the early postoperative stage following spinal implantation [23].

In order to reduce the amount of computation to minimize computation time, we utilized unilateral fixation (ISES-UF) in all experiments, discussing the optimal entry point of the screws. The ISES-1/2 and ISES-1/3 models were established by inserting screws from the cephalad one-half and one-third of the spinolaminar junction, respectively [12], passing the length of the lamina and ending near the facet joints, while maintaining the integrity of the facet joints. In both models, the screws were constrained by an elastic rod. The material of the interlaminar screw and the shell of elastic rod was Ti-6Al-4V alloy, and the Young’s modulus and Poisson’s ratio of the model were set to 113 GPa and 0.3 in Abaqus 2021. The material of the nickel–titanium ‘cable’ structure was nickel–titanium, and the Young’s modulus and Poisson’s ratio of the model were set to 47 GPa and 0.3 in Abaqus 2021.

In the ISES-BF model, one screw was inserted from the cephalad one-half spinous junction and the other screw was inserted from the cephalad one-third spinous junction. Due to the limited space within the lamina, bifurcation of the implanted screws prevents collision between the two screws or fracture of the lamina.

In the Coflex-F model, a portion of the interspinous process and interspinous ligament were resected to provide adequate space for implanting the Coflex-F device between the interspinous process of the lumbar L4/L5. The rivets were simplified as cylinders and constrained to all degrees of freedom on the holes in the Coflex-F’s wings and the spinous processes [23]. The surface between the U-shaped structure of the Coflex-F device and the spinous process was modeled as surface-to-surface contact with a coefficient of friction of 0.8, and the coefficient of friction for the rest of the contact regions was set to 0.1 [23]. The Young’s modulus and Poisson’s ratio of the Coflex-F were assigned to be 113 GPa and 0.3, respectively [23].

### 2.3. Simulation Analysis

The bottom of the L5 segment was constrained in all directions, and the compressive force of 300 N and the moments of 7.5 Nm were applied to the upper surface of the L4 segment to simulate the general physiological state of flexion, extension, lateral bending, and axial rotation, respectively [17]. Since this study contains asymmetric fixation (ISES-UF), flexion and rotation of the right and left sides are specifically discussed. To analyze the optimal insertion point for the ISES technique, the biomechanical parameters of INT, ISES-1/2, and ISES-1/3 models were compared after applying forces and moments. The more dominant insertion point in the ISES-1/2 model and ISES-1/3 model was considered as the dominant insertion point and applied in the later discussion, and the corresponding model was defined as the ISES-UF model. Unilateral fixation has smaller wounds and faster recovery. To investigate the feasibility of unilateral fixation in the ISES technique, we compared the biomechanical parameters of the ISES-UF model and ISES-BF model. The more dominant of the ISES-UF model and ISES-BF model was defined as the ISES model and compared with the Coflex-F model [23].

Biomechanical parameters include the IDP, the ROM, the average von Mises stress at the facet cartilage (FCS), and the posterior structure (PSS), which predict the risk of various complications.

## 3. Results

### 3.1. Validation of the Intact Lumbar FE Model

The average pressure under the compression of 150 N, 400 N, and 1000 N was 0.147 MPa, 0.419 MPa, and 1.049 MPa, respectively, which were within the range of the experimental data [16,19]. The ROMs under the compression of 300 N and moments of 3 Nm, 7.5 Nm, and 10 Nm all fall within the range of the experimental data [20,21,22], indicating that the model is reliable. The data for the details used for validation have been plotted (see Appendix A).

### 3.2. The Insertion Point of ISES Technique

As shown in Figure 2 and Figure 3, compared with the ISES-1/2 model of the lumbar spine, the IDP values of the ISES-1/3 model were slightly larger in extension, lateral bending, and axial rotation. In particular, the IDP in the ISES-1/2 model was 0.15 MPa in extension. This suggests that ISES-1/2 may provide better relief for IDP. This could be related to the angle of the screws, which can better distribute the load on the lumbar spine.

The differences of mobility between the ISES-1/3 model and the ISES-1/2 model were not significant in flexion, lateral bending, and rotation. In extension, the ISES-1/3 model exhibits significantly higher mobility compared to the ISES-1/2 model. However, for patients with a degenerative lumbar spine, excessive lumbar extension can lead to the narrowing of the spinal canal and a reduced neural foraminal area. Therefore, given that the ISES-1/2 and ISES-1/3 models have a relatively small impact on flexion, lateral bending, and rotation, we believe that the clinical effectiveness of the ISES-1/2 model is superior.

As shown in Figure 2 and Figure 4, the FCS of the ISES-1/2 model was less than that of the ISES-1/3 model in flexion, extension, lateral bending, and axial rotation. During extension, the ISES-1/3 model exhibits a greater FCS. This is likely because the ISES-1/3 model has greater mobility, transferring more load onto the facet joints during lumbar spine extension. Additionally, due to the unilateral fixation approach employed by the ISES technique, the FCS of the model is not symmetrical during left and right bending as well as left and right rotation. The maximum stress values in the posterior structure of both the ISES-1/2 model were smaller than those of the ISES-1/3 model in flexion, lateral bending, and axial rotation. The location of the maximum stress in the posterior structure of the ISES-1/2 model and the ISES-1/3 model was located at the interlaminar screw entry point.

Based on the above results, it is recommended that the midpoint of the lumbar lamina (ISES-1/2) is the optimal insertion point for inserting the interlaminar screw with the lower von Mises stress of the facet cartilage and posterior structure. The ISES-1/2 model can effectively reduce the IDP and showed a slight advantage to the ISES-1/3 model in reducing the stress on the facet cartilage and the stress on the posterior structure.

### 3.3. Feasibility Analysis of the Unilateral Fixation in the ISES Technique

As shown in Figure 3 and Figure 5, the IDP values of the ISES-BF model were slightly larger than those of the ISES-UF (ISES-1/2) model in flexion, extension, lateral bending, and axial rotation, but the differences were not significant. This indicates that increasing the quantity of internal fixation does not better alleviate IDP.

The mobility of the ISES-UF model was greater than that of the ISES-BF model in flexion, extension, lateral bending, and axial rotation. The results indicate that compared to unilateral fixation, bilateral fixation reduces the mobility of the lumbar spine.

As shown in Figure 4 and Figure 5, the FCS of the ISES-UF model was larger than that of the ISES-BF model in extension, left bending, and right rotation. The differences of FCS between the ISES-UF model and the ISES-BF model were not significant in flexion, extension, left lateral bending, and rotation. However, during right bending, the FCS in the ISES-UF model is significantly higher than that in the ISES-BF model. This may be due to stress concentration resulting from unilateral fixation. Nevertheless, during right bending, the FCS in the INT model is 1.67 MPa, and the FCS in the ISES-UF model is 1.44 MPa, indicating that the FCS in the ISES-UF model remains lower than that in the INT model.

The PSS of the ISES-UF model was smaller than those of the ISES-BF model in flexion, lateral bending, and axial rotation. The location of the maximum stress in the posterior structure of the ISES-BF model was located at the interlaminar screw entry point. The results indicate that adding additional internal fixation increases the risk of posterior structure fracture.

Based on the above results, while unilateral fixation with ISES technology may potentially increase FCS, it remains effective in reducing IDP, providing a greater ROM, and resulting in a smaller PSS. Therefore, we believe that unilateral fixation with ISES technology holds advantages over bilateral fixation.

### 3.4. The Feasibility of ISES Technique

As shown in Figure 3 and Figure 6, the ISES (ISES-1/2) model had a slightly lower IDP in extension, lateral bending, and axial rotation than the Coflex-F model, and a slightly higher IDP in flexion, but the difference in the IDP values between the two models were not significant. The results demonstrate that the ISES model has a better effect in alleviating the IDP.

The ISES model showed a greater ROM than the Coflex-F model in flexion, extension, and lateral bending. There was little difference in mobility between the two models under right axial rotation and left axial rotation. Compared to the Coflex-F device, the elastic rods in the ISES model provided greater lumbar spine mobility.

As shown in Figure 4 and Figure 6, the FCS of the ISES model in flexion was not significantly different from those of the Coflex-F model. The FCS of the ISES model in extension, left lateral bending, and axial rotation were smaller than those of the Coflex-F model. The FCS of the Coflex-F model was 1.51 MPa under right axial rotation and 1.46 MPa under left axial rotation. In the Coflex-F model, the right axial rotation FCS was 1.51 MPa and the left axial rotation FCS was 1.46 MPa, while in the ISES model, the right axial rotation FCS was 0.52 MPa and the left axial rotation FCS was 0.58 MPa. This may be due to the fact that the semi-rigid fixation in the Coflex-F device does not effectively distribute the load on the facet joint in rotation, while the elastic rods in the ISES model better distribute this stress.

The maximum von Mises stress on the posterior structure in the Coflex-F model was at the contact of the rivets with the posterior structure. The PSSs of the ISES-1/2 model were significantly smaller than those of the Coflex-F model. Especially in flexion, the constraints imposed by the rivets in the Coflex-F device can easily lead to spinous process fractures, whereas the elastic rods in the ISES technique provide excellent mobility in lumbar flexion, dispersing stress on the posterior structures.

## 4. Discussion

Lumbar degenerative diseases are a debilitating condition associated with the degeneration of the spine that occurs with age. The ISES technique is a novel interlaminar device that consists of interlaminar screws and elastic rods. However, as a new technology, there are uncertainties regarding the optimal screw entry point, whether the fixation should be unilateral or bilateral, and whether this new technology is superior to the currently employed methods. Therefore, the purpose of our study is to investigate the feasibility of the ISES technique for treating lumbar degenerative diseases from a biomechanical perspective. Our study found that the midpoint of the spinolaminar junction is the optimal insertion point for interlaminar screws. We have also verified that unilateral fixation yields comparable stability to bilateral fixation. Furthermore, compared with the Coflex-F model, the ISES model is as effective as the Coflex-F model in IDP and limiting extension, retains more of a range of motion, and reduces the FCS and PSS. The ISES technique has the potential to bring new breakthroughs in the treatment of lumbar degenerative diseases and contribute to the rehabilitation and improvement of the quality of life for patients, especially those with osteoporosis.

For patients with degenerative lumbar spine disease, common treatment approaches can be categorized into two main categories: conservative treatment and surgical treatment. Conservative treatment includes physical therapy and medication management. However, the effectiveness of conservative treatment is influenced by the subjective experiences of the patients. Research by Enrico Ballestra et al. suggested that the effectiveness of conservative treatment in patients is correlated with the patients’ subjective experiences during the treatment process [24]. In addition, radiofrequency, as a minimally invasive surgical technique, has been proven to provide significant relief from lower back pain caused by degenerative lumbar conditions [25,26]. However, for patients with surgical indications, surgical treatment remains the most effective method for alleviating symptoms of degenerative lumbar diseases.

While internal fixation and fusion has been widely accepted as the standard treatment method for degenerative lumbar spine disease [27], there is still controversy surrounding the choice between fusion surgery and non-fusion surgery. A literature review and meta-analysis study’s results indicate that posterior lumbar non-fusion surgery is superior to posterior lumbar interbody fusion in terms of reducing surgical time, minimizing intraoperative bleeding, preserving the ROM of surgical segments, and preventing adjacent vertebral segment degeneration to some extent [28]. A finite element analysis comparing the biomechanics of a posterior dynamic fixation device with anterior interbody lumbar fusion under whole-body vibration conditions showed that the dynamic fixation device model exhibited a lower amount of stress on the endplate and screws at the fixed segment compared to fusion [29]. Lumbar spine dynamic fixation techniques include pedicle screw-based dynamic stabilization devices, interspinous spacers, and facet joint replacement surgery [30,31]. Among these, one of the most widely used techniques is the use of interspinous spacers. The ISP techniques, such as Coflex, Wallis, X-STOP, and Bacfuse, utilize distinct designs and application methods [32,33]. Although their clinical outcomes differ, the core principle behind these technologies is to exert pressure on the spinous processes to limit lumbar extension. However, because the spinous processes are inherently fragile, this frequently leads to complications such as spinous process fractures, implant subsidence, and implant migration. In contrast, the ISES technique employs a screw–rod system that distributes the force to the lamina, while adjusting the distance of the screws to stabilize the space between the posterior structures of the lumbar spine. Trautwein et al. showed that the disruption strength of the lamina is 2–5 times higher than that of the spinous process [7]. Therefore, the ISES technique not only provides superior implant stability, but also reduces the risk of implant displacement. The ISES technique also preserves the integrity of the spinous process and the surrounding ligaments in spine surgery, thereby facilitating the delivery of posterior loads more effectively.

Unsafe screw placement is associated with a risk of injury to the blood vessels, spinal cord, and nerves. Therefore, this study compared two well-accepted methods for screw placement with reference to the insertion point in the TT technique [12,13,14]. In this study, the ISES-1/2 model was found to be effective in relieving the IDP in the lumbar spine, providing greater mobility in flexion, limiting the extension of the lumbar spine, having a lower than average FCS, and having less of a PSS. The reduced mechanical loading on the posterior bone (PSS) can lead to a decrease in bone density, which, over time, may result in bone resorption or bone loss around the implant. Stress shielding is often considered significant when there is a reduction in physiological mechanical loading on the bone by approximately 30% or more due to the presence of an implant [34]. However, in our study, the distribution of PPS was homogeneous across the groups, and none of the groups had a PPS reduction of more than 30%, so the risk of stress shielding was low. Hu et al. elucidated the trend from superior to inferior in the initial, median, and distal regions of the lamina by anatomical measurements. They concluded that the optimal screw entry point is located at the median of the base of the posterior structure [14]. This finding is consistent with the results of our study. Additionally, Singhatanadgige et al. found that the angle between the screw trajectory and the perpendicular line of the spinolamina of the ISES-1/3 technique was greater than that of the ISES-1/2 technique [12]. Considering that a smaller angle facilitates screw insertion through the mini-open incision [35], the ISES-1/2 technique is more recommended. Therefore, selecting the position of the cephalad one-half of the spinolaminar junction as the insertion point for the ISES technique is more sensible.

Bilateral fixation is commonly employed in conventional surgery, which provides good stability of the spine. However, bilateral fixation is associated with problems such as rigid fixation, significant intraoperative blood loss, and higher costs [36]. Several studies have shown that the unilateral fixation of the lumbar spine offers advantages such as smaller surgical incisions, faster recovery, and the preservation of spinal mobility, but the feasibility of unilateral fixation is currently controversial [37,38,39]. The goal of the ISES technique is to achieve the increased preservation of spinal mobility in the patients, while utilizing minimally invasive surgery. Therefore, we advocate the utilization of unilateral fixation. However, further investigation is required to determine whether unilateral fixation offers sufficient stability. This study compared the ISES-UF model with the ISES-BF model. The results showed that ISES-UF exhibits a reduced IDP and PSS, as well as improved flexibility and decreased FCS in both flexion and extension. These findings imply that unilateral fixation is less effective in reducing the risk of lumbar facet joint hypertrophy and vertebral lamina fractures, and decreasing the degeneration of adjacent vertebral segments, among other factors. Additionally, paraspinal muscles play a crucial role in maintaining spinal stability, and unilateral fixation minimizes the damage to these muscles [40,41]. Regarding the ISES technique, unilateral fixation provides comparable stability to bilateral fixation and postpones the progression of lumbar degenerative diseases. Therefore, we believe that the ISES technique using unilateral fixation is feasible and has certain advantages.

After determining the optimal entry point and fixation of the ISES technique, the mechanical characteristics of the ISES technique were compared with those of the Coflex rivet technique. The ISES technique restricts lumbar extension by bracing the upper and lower posterior structures of the lumbar spine with a screw–rod system. This technique effectively limits lumbar extension while preserving greater mobility. This is mainly due to the structural design of the novel elastic rod, which further restricts excessive flexion and helps avoid lumbar spine instability. Arthritis of the facet joint may result in loads as high as 47% [42] and accelerates the degeneration of the facet joint [43]. This degeneration typically initiates with changes in the articular cartilage and eventually leads to the complete destruction of the facet joint. Both the ISES technique and the Coflex-F technique contribute to stress reduction in the facet joint, thus lowering the risk of stress-related facet joint pain, and ultimately improving the quality of life for patients post-surgery.

There is also a potential risk of screw displacement and pullout with the ISES technique during long-term lumbar spine activities, especially for patients with osteoporosis. Due to the aging population, it is expected that the number of osteoporosis patients will significantly increase in the next 10 years [44,45]. Therefore, in order to reduce the risk of screw displacement, the orientation of the screw in the ISES technique maximizes the contact area between the screw and the bone tissue structure of the vertebral lamina. This enhances the screw’s resistance to extraction and reduces the risk of interlaminar screw pullout. Furthermore, the ISES technique uses larger diameter screws to reduce the risk of screw loosening and displacement. Previous studies showed that the outer diameter of the lamina gradually decreased from 7.18 to 6.59 mm for L1–L5, and the use of 4.5 mm diameter screws was recommended for L3–L5 [12]. Thus, this study used interlaminar screws with a diameter of 4.5 mm, slightly larger than those used in the TT technique [46,47], to enhance the pullout resistance of the screws. Excellent screw resistance to pullout provides an opportunity for surgery in patients with osteoporosis. Severe osteoporosis is an absolute contraindication to interspinous spacer implantation, as fractures may occur if the spine is mismatched with the implant during or after surgery. Combined muscle strengthening exercises have been shown to improve bone density in the lumbar vertebral body [48], so patients can be advised to strengthen their muscles appropriately. The ISES technique preserves the integrity of the lumbar spine, and when the interlaminar screws are withdrawn, follow-up treatment allows for the reimplantation of the lumbar interlaminar screws and rod on the other side, and also allows for lumbar fusion surgery treatment, significantly delaying the fusion procedure.

There are some limitations to this study. First, this was just a preliminary assessment of the reliability of the ISES technique for the treatment of degenerative lumbar spine disease. Therefore, our study focuses on comparing various non-fusion models and does not involve a comparison between the ISES technique and fusion technique. After proving the biomechanical feasibility of the ISES technique, further validation by animal models and clinical trials will be required in the future. Second, the musculature and several degenerative parameters, such as ligamentous laxity, were not considered in the lumbar FEM, which may have affected the stability and flexibility of the spine. Paraspinal muscles play a crucial role in maintaining the stability of the lumbar spine. However, the effects of the muscle on the lumbar spine are complex, and there are no authoritative data that can be used as a reference for simulation and analysis. Our research primarily centers on a comparative analysis of various aspects, including different insertion points, fixation methods, and surgical techniques. Therefore, it is sufficient to ensure that all conditions are the same in each group except for the factor being compared, and it is not essential to include the role of muscle. Third, because L4/5 is the lumbar spine segment with the highest incidence of degenerative lumbar spine disease, only the feasibility of the ISES technique for L4/5 was considered in this study. The ISES technique may have different biomechanical results in different lumbar segments. Fourth, this study did not consider parameters such as screw length, diameter, and thread that are equally important for the stability of the fixation system. This study was to determine how the ISES technique operates (including the choice of nail placement points and fixation rods) and to evaluate the biomechanics of the ISES technique. The geometric parameters of the screw can be considered in future studies to optimize the ISES technique. Finally, the FE model did not directly assess the impact of ISES technology on the adjacent vertebral segment, but instead used the axial stiffness of the fixed vertebrae to predict the risk of adjacent vertebral diseases.

## 5. Conclusions

The results of our study suggest that the ISES technique can effectively relieve intervertebral pressure, preserve grater mobility, and reduce the FCS and PSS. Therefore, our study provided the initial validation of the feasibility of using ISES technology in the treatment of lumbar degenerative diseases, especially for those with osteoporosis.

## Figures and Tables

**Figure 1 bioengineering-10-01204-f001:**
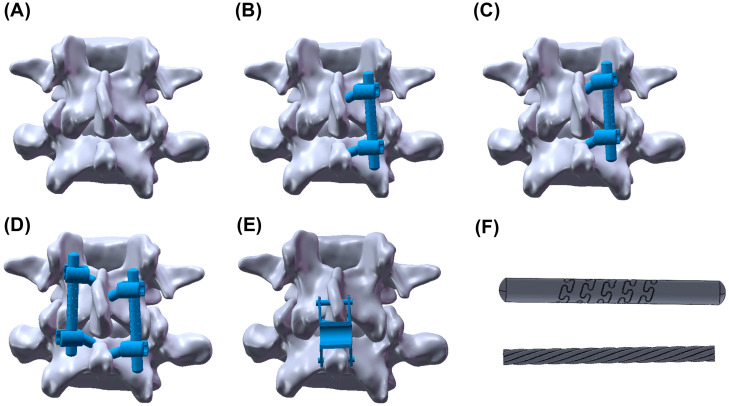
The intact lumbar spine model (**A**), the ISES-1/2 model (**B**), the ISES-1/3 model (**C**), the ISES-BF model (**D**), the Coflex-F model (**E**), and the elastic rod (**F**).

**Figure 2 bioengineering-10-01204-f002:**
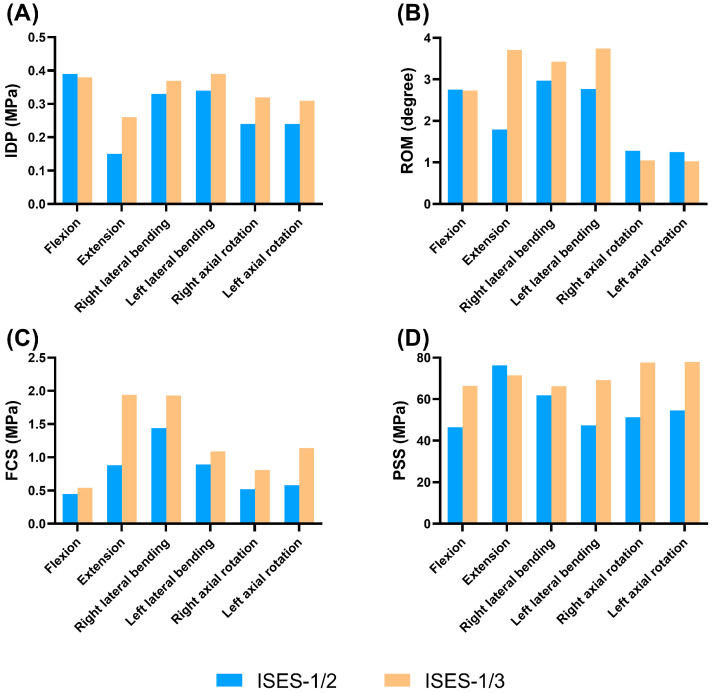
Parameters of the ISES-1/2 model (blue bar) and ISES-1/3 model (orange bar) under flexion, extension, lateral bending, and axial rotation. (**A**) The IDP, (**B**) ROM, (**C**) the mean stress of facet cartilage, and (**D**) the maximum stress of posterior structure.

**Figure 3 bioengineering-10-01204-f003:**
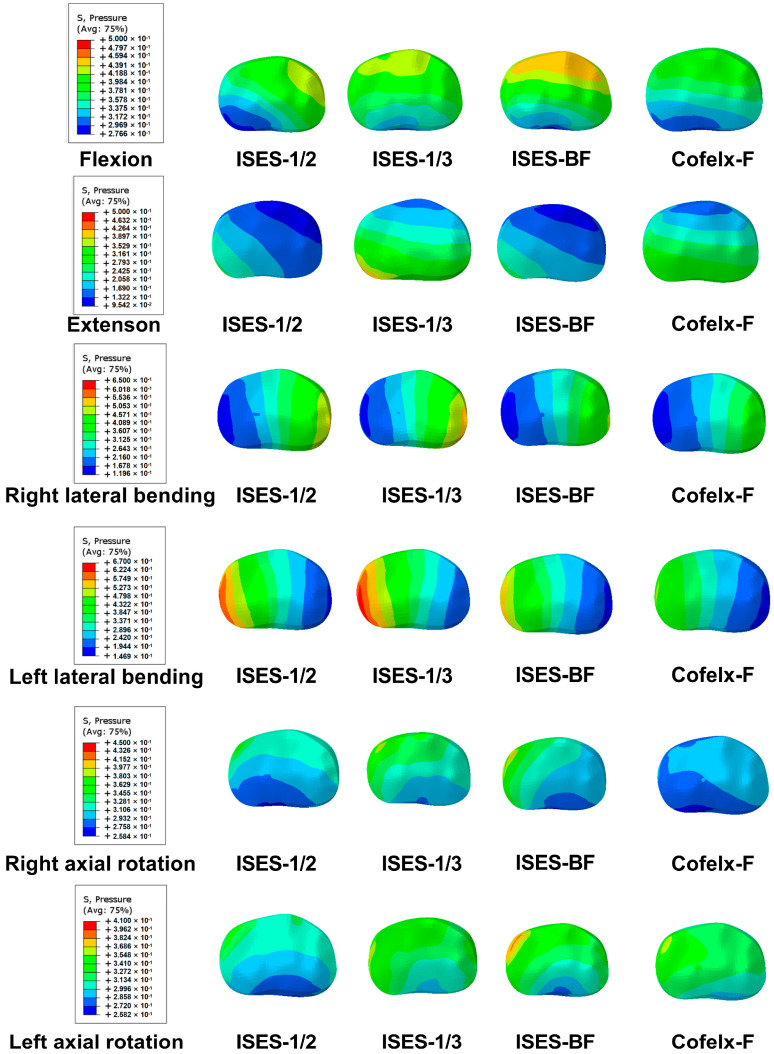
The intradiscal pressure (IDP) of the ISES−1/2 model, ISES−1/3 model, ISES−BF model, and Coflex−F model under flexion, extension, lateral bending, and axial rotation. Red represents large values and blue represents small values. The color bar on the left shows the value of the detail.

**Figure 4 bioengineering-10-01204-f004:**
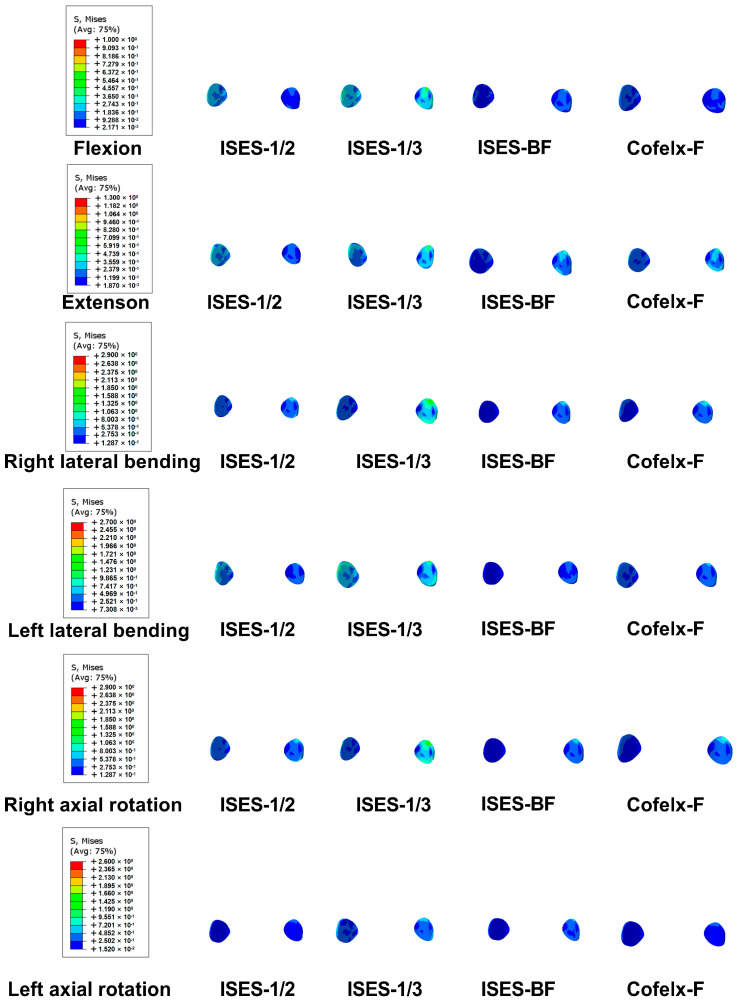
The average von Mises stress at the facet cartilage (FCS) of the ISES−1/2 model, ISES−1/3 model, ISES−BF model, and Coflex−F model under flexion, extension, lateral bending, and axial rotation. Red represents large values and blue represents small values. The color bar on the left shows the value of the detail.

**Figure 5 bioengineering-10-01204-f005:**
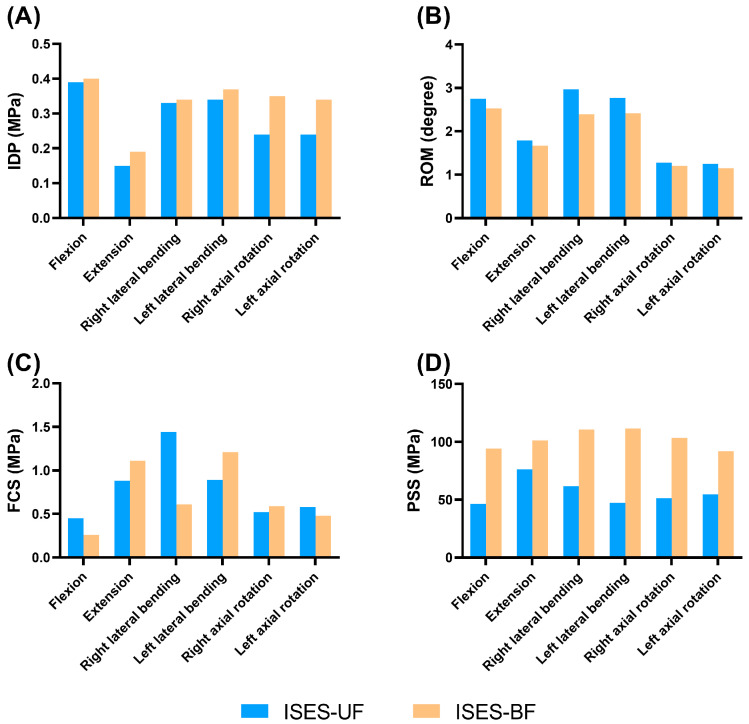
Parameters of the ISES-UF model (blue bar) and ISES-BF model (orange bar) under flexion, extension, lateral bending, and axial rotation. (**A**) The IDP, (**B**) ROM, (**C**) the mean stress of facet cartilage (FCS), and (**D**) the maximum stress of posterior structure (PPS).

**Figure 6 bioengineering-10-01204-f006:**
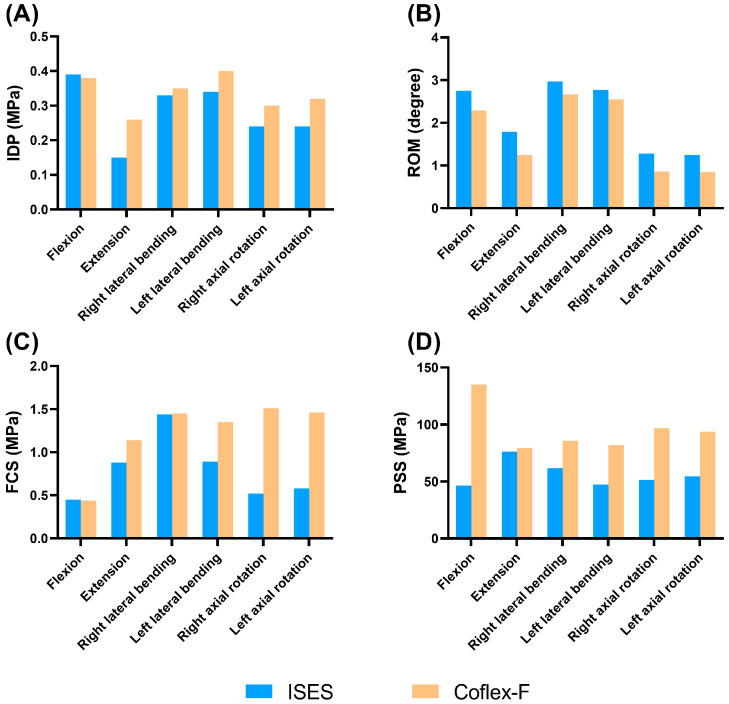
Parameters of the ISES model (blue bar) and Coflex-F model (orange bar) under flexion, extension, lateral bending, and axial rotation. (**A**) The IDP, (**B**) ROM, (**C**) the mean stress of facet cartilage (FCS), and (**D**) the Maximum stress of posterior structure (PPS).

**Table 1 bioengineering-10-01204-t001:** Properties of different components in the INT model.

Components	Young’s Modulus (MPa)	Poisson’s Ratio	Cross-Sectional Area (mm^2^)	Type Element
Cortical bone	12,000	0.3	-	T3D8
Cancellous bone	100	0.2	-	T3D8
Posterior bone	3500	0.25	-	T3D4
Endplate	500	0.3	-	T3D4
Nucleus pulposus	1	0.49	-	T3D8
Annulus fibrosus	4.2	0.45	-	T3D8
Annulus fiber layers	360–550	-	0.76	T3D2
ALL	15.6–20	0.3	63.7	T3D2
PLL	10–20	0.3	18	T3D2
LF	13–19.5	0.3	40	T3D2
CL	7.5–33	0.3	32	T3D2
ITL	12.0–58.7	0.3	1.8	T3D2
ISL	8.8–15	0.3	25.2	T3D2
SSL	9.8–12	0.3	35.1	T3D2
Screw	113,000	0.3	-	T3D8
The shell of elastic rod	113,000	0.3	-	T3D4
The ‘cable’ structure	47,000	0.3	-	T3D4

## Data Availability

The raw data supporting the conclusion of this article will be made available by the authors, without undue reservation.

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
