# Peer review of "Treatment of Lumbar Degenerative Disease with a Novel Interlaminar Screw Elastic Spacer Technique: A Finite Element Analysis"

_bioengineering, 2023, doi:10.3390/bioengineering10101204_

Round 1
Reviewer 1 Report
In this manuscript, the authors constructed a FE model and calculated the stresses and the range of motion (ROM) of spinal segments to check that the interlaminar elastic screw spacer technique can slow the progression of lumbar degenerative disease with few complications.
The article is of low quality for the following reasons:
ABSTRACT: Abbreviations are not appropriate in this section.
INTRODUCTION: In this section, you referenced 14 citations; approximately 57% are over ten years old. Many references are too outdated. As it is old information, there is no relevance.
MATERIALS AND METHODS: This section should begin by stating the methodological design of the finite element method. Also, you can summarize the mechanical properties of your system in a table. I am more concerned about establishing finite element models, including necessary structures such as ligaments. In addition, detailed model meshing reconstruction, the element used, boundaries, and loading conditions must be introduced (Show a picture remaking the applied load´s and boundary condition´s zone). You must mention the vertebrae´s assumption and their mechanical properties. Add all the necessary information related to the intervertebral disc. Finally, briefly explain the reason for the applied loads.
RESULTS: This section is very weak. You must include model validation and a mesh analysis. Also, you should explain the behavior of the model. For example, in Figure 3, why is the pressure distribution higher in ISES-1/3 in left lateral bending than in the others?
DISCUSSION: This section would also benefit from considering that the muscles are a significant factor in lumbar activity, and this article did not consider them. It must be discussed. You must explain and compare your simulation results against the literature’s information. Also, you must add a stress shielding analysis and how the different stiffness could affect the patient's recovery.
The legends of Figures and Tables are too simple and require a more specific introduction.
The manuscript´s structure does not have any methodology, so there is no repeatability.
The limitations of the research have been underestimated. Authors need to be more honest.
no comments
Author Response
We would like to extend our sincere gratitude for taking the time to review our manuscript. Your feedback and insights are invaluable in enhancing the quality and comprehensiveness of our research. We appreciate your thoughtful suggestion and I will respond to your suggestions one by one.
ABSTRACT: Abbreviations are not appropriate in this section.
A1: Thank you for your recommendation, we have thoroughly reviewed the designated section and have made the necessary revisions to ensure that no acronyms are utilized therein. We would like to express our gratitude for your constructive input, which undoubtedly enhances the overall quality of our research article.
INTRODUCTION: In this section, you referenced 14 citations; approximately 57% are over ten years old. Many references are too outdated. As it is old information, there is no relevance.
A2: We would like to express our gratitude for highlighting the concern regarding the age of some of our cited references. In response to your valuable feedback, we have meticulously reevaluated and updated the references in the specified section, specifically, we have replaced the original citations [1]-[5] and [10] with [3]- [6] and [11]. We have replaced the outdated citations with more recent and relevant sources that better align with the current state of research in the field. This ensures that our readers have access to the most up-to-date and pertinent information, enhancing the scholarly value of our work.
We understand the critical importance of citing recent and relevant sources to support our arguments and conclusions. Your suggestion has helped us improve the rigor and currency of our manuscript, and we are genuinely grateful for your guidance in this regard.
We have removed the following references:
- Sobottke, R., et. al. Interspinous implants (X Stop, Wallis, Diam) for the treatment of LSS: is there a correlation between radiological parameters and clinical outcome? European spine journal : official publication of the European Spine Society, the European Spinal Deformity Society, and the European Section of the Cervical Spine Research Society 2009, 18, 1494-1503.
- Kabir, S., et. al. Lumbar interspinous spacers: a systematic review of clinical and biomechanical evidence. Spine 2010, 35, E1499-1506.
- Wilke, H., et. al. Testing criteria for spinal implants: recommendations for the standardization of in vitro stability testing of spinal implants. European spine journal : official publication of the European Spine Society, the European Spinal Deformity Society, and the European Section of the Cervical Spine Research Society 1998, 7, 148-154.
- Barbagallo, G.et. al.Analysis of complications in patients treated with the X-Stop Interspinous Process Decompression System: proposal for a novel anatomic scoring system for patient selection and review of the literature. Neurosurgery 2009, 65, 111-119; discussion 119-120.
- Xu, C.et. al. Complications in degenerative lumbar disease treated with a dynamic interspinous spacer (Coflex). International orthopaedics 2013, 37, 2199-2204.
- Ren, C.et. al. Adjacent segment degeneration and disease after lumbar fusion compared with motion-preserving procedures: a meta-analysis. European journal of orthopaedic surgery & traumatology : orthopedie traumatologie 2014, S245-253.
And replace them with new references:
- Raikar, Set. al. Inter Spinal Fixation and Stabilization Device for Lumbar Radiculopathy and Back Pain. Cureus 2021, 13, e19956.
- Faulkner, J.et. al. Interspinous Process (ISP) Devices in Comparison to the Use of Traditional Posterior Spinal Instrumentation. Cureus 2021, 13, e13886.
- Wei, H.et. al. Preliminary efficacy of inter-spinal distraction fusion which is a new technique for lumbar disc herniation. International orthopaedics 2019, 43, 899-907.
- Li, K.et. al. Complications and radiographic changes after implantation of interspinous process devices: average eight-year follow-up. BMC musculoskeletal disorders 2023, 24, 667.
- Oh, H.et. al.. The Relationship between Adjacent Segment Pathology and Facet Joint Violation by Pedicle Screw after Posterior Lumbar Instrumentation Surgery. Journal of clinical medicine 2021, 10.
MATERIALS AND METHODS: This section should begin by stating the methodological design of the finite element method. Also, you can summarize the mechanical properties of your system in a table. I am more concerned about establishing finite element models, including necessary structures such as ligaments. In addition, detailed model meshing reconstruction, the element used, boundaries, and loading conditions must be introduced (Show a picture remaking the applied load´s and boundary condition´s zone). You must mention the vertebrae´s assumption and their mechanical properties. Add all the necessary information related to the intervertebral disc. Finally, briefly explain the reason for the applied loads.
A3: Your feedback has been immensely valuable in refining the methodology section of our research, particularly with regard to the finite element model. Since this finite element model is described in detail in another paper we have published, we have omitted many details in this manuscript. We have incorporated the changes you suggested to enhance the clarity and comprehensibility of our methods.
In response to your recommendations, we have introduced a dedicated subsection 2.1 in the Methods section that provides a comprehensive overview of our finite element model. This new subsection elucidates the methodological design of the finite element method, including essential structural components such as the necessary structure, cell types, boundaries, and loading conditions. Moreover, we have created a table (Table 1) that succinctly summarizes the material properties of our model, ensuring that readers can readily access this critical information. The loads in this study are all set up with reference to the published literature and are modeled to be the loads that a person would experience under general physiological states. We have also added a brief explanation of the reasons for applying loads in the line 164 of the manuscript.
RESULTS: This section is very weak. You must include model validation and a mesh analysis. Also, you should explain the behavior of the model. For example, in Figure 3, why is the pressure distribution higher in ISES-1/3 in left lateral bending than in the others?
A4: In response to your thoughtful suggestions, we have taken significant steps to enhance the comprehensiveness and clarity of our results section. We have structured this section into 4 distinct subsections, aligning with the format of the Methods section, to facilitate a more systematic and in-depth presentation of our findings.
The mesh analysis have been added in the Methods section (Lines 109-111). In subsection 3.1, we have added the results of the validation of the model. Given the volume of data, we have employed graphical representation to convey the information effectively. This graph was used in a previously published article, so we present it as supplementary. In subsection 3.2 we presented the results for different insertion points.In subsection 3.3 you presented the results for the ISES technique with unilateral and bilateral fixation.In subsection 3.4 we presented the results for the ISES technique and the Coflex-F technique. However we did not include too much explanation of the results in this section, but put the explanation of the results in the Discussion section.
DISCUSSION: This section would also benefit from considering that the muscles are a significant factor in lumbar activity, and this article did not consider them. It must be discussed. You must explain and compare your simulation results against the literature’s information. Also, you must add a stress shielding analysis and how the different stiffness could affect the patient's recovery.
A5: While we acknowledge that the musculature around the lumbar spine plays an essential role in spine health, our research did not encompass an analysis of muscle-related factors. As such, we are unable to provide empirical data or analysis on this topic. In the Limitations section of our manuscript, we will make explicit note of the absence of muscle-related analysis.
The effects of muscle on the lumbar spine are complex, and there is no authoritative data that can be used as a reference for simulation and analysis. Our research primarily centers on a comparative analysis of various aspects, including different insertion points, fixation methods, and surgical techniques. Therefore it is sufficient to ensure that all conditions are the same in each group except for the factor being compared, and it is not essential to include the role of muscle.
Nonetheless, we firmly believe that our study makes a valuable contribution to the field of spinal surgery. It provides clinicians and researchers with proof of the feasibility of surgical techniques in which ISES technology is effective in improving the lives of patients, which contributes to improved surgical techniques, implant design, and more. While the current study may not delve into the muscle-related aspects of lumbar spine health, we acknowledge the importance of this topic. We see it as a potential avenue for future research, and we are committed to exploring it in subsequent studies, where we can allocate the necessary resources and focus to address this crucial dimension comprehensively.
As per the existing literature, the risk of developing stress shielding becomes notably significant when there is a reduction in supraskeletal stress by more than 30%. Our analysis revealed that the PPS distribution across all groups is uniform. None of the groups exhibited a reduction in supraskeletal stress exceeding the 30% threshold mentioned in the literature. Based on these findings, we have ascertained that the risk of stress shielding is low in our study. This information has been incorporated into lines 347-353 of the manuscript to underscore our consideration of the stress shielding risk.
The legends of Figures and Tables are too simple and require a more specific introduction.
A: We acknowledge the importance of providing clear and informative legends to enhance the understanding of our visual representations. In response to your valuable suggestion, we have taken steps to augment the legends with more specific and elucidating introductions. For example, we added a description of the colors in the graph. In addition, we split the originally complex 1 diagram (Figure 2) into 3 diagrams (Figures 2-4), which is more conducive to the reader's understanding of the diagrams.
The manuscript´s structure does not have any methodology, so there is no repeatability.
A: Wehave undertaken significant revisions to both the Methods and Results sections of the manuscript. We have made a dedicated effort to provide meticulous and comprehensive details of the manipulations and procedures carried out during the course of the study. These enhancements aim to offer readers a clear and replicable roadmap, ensuring that our research can be reproduced with precision and accuracy.
The limitations of the research have been underestimated. Authors need to be more honest.
A: We genuinely appreciate your critical assessment of our manuscript and your emphasis on the importance of transparently acknowledging the limitations of our study. We wholeheartedly agree that there are several areas in which our study could be improved; however, we must acknowledge that accomplishing these enhancements is beyond the scope of our current research endeavor. Furthermore, in response to your guidance, we have introduced two additional limitations to the manuscript, which we believe more accurately convey the boundaries of our study. These specific changes are now reflected in lines 418-440 of the manuscript.
We would like to express our sincere gratitude for your valuable feedback and your diligent evaluation of our manuscript. Your insights have been pivotal in enhancing the quality and reproducibility of our study. Once again, We are sincerely grateful for your review and your dedication to advancing the standards of scholarly research.

Reviewer 2 Report
Dear Sirs,
Your paper seems interesting but some concerns have to be better addressed.
lt is not clear the study design: is it an observational study? You should clearly state it both in the abstract and in the materials and methods section.
Moreover, Your study needs some clinical concerns. Did you involve patients in this research? How is possible to consider the clinical improvement you stated without clinical applications?
The discussion should be briefly integrated giving your results some clinical lapels. To do that, I suggest the following references:
Farì, G., de Sire, A., Fallea, C., Albano, M., Grossi, G., Bettoni, E., Di Paolo, S., Agostini, F., Bernetti, A., Puntillo, F., & Mariconda, C. (2022). Efficacy of Radiofrequency as Therapy and Diagnostic Support in the Management of Musculoskeletal Pain: A Systematic Review and Meta-Analysis. Diagnostics (Basel, Switzerland), 12(3), 600. https://doi.org/10.3390/diagnostics12030600
Giglio, M., Farì, G., Preziosa, A., Corriero, A., Grasso, S., Varrassi, G., & Puntillo, F. (2023). Low Back Pain and Radiofrequency Denervation of Facet Joint: Beyond Pain Control-A Video Recording. Pain and therapy, 12(3), 879–884. https://doi.org/10.1007/s40122-023-00489-y
Bets regards
Author Response
Thank you for your valuable feedback on our paper. We appreciate your thoughtful comments and suggestions, and we are committed to addressing them in order to improve the clarity and quality of our research. Below, we provide responses to your concerns and recommendations:
lt is not clear the study design: is it an observational study? You should clearly state it both in the abstract and in the materials and methods section.
A: We deeply appreciate your thoughtful review. Our research employs finite element simulation analysis, which is an engineering simulation technique widely used to model and analyze the behaviors of physical systems or structures. It is primarily employed for predictive purposes, enabling us to assess various aspects of the system's performance, strength, stability, and other pertinent properties, ultimately saving time and resources
Given the nature of our research, it is essential to emphasize that it falls within the domain of computational engineering rather than being an observational study. Our study design is centered on the creation of a robust computational model that accurately represents the physical system under investigation. This model is then subjected to controlled simulations to analyze and predict its behavior under various conditions. We have expressly articulated this study design in lines 81-84 of our manuscript.
Moreover, Your study needs some clinical concerns. Did you involve patients in this research? How is possible to consider the clinical improvement you stated without clinical applications?
A: Our research did not directly involve patients but was designed as a simulation to predict clinical improvement.
Finite element analysis is a numerical simulation technique that allows researchers to model complex physical processes, assess design performance, optimize systems, and conduct virtual experiments. Through finite element analysis, we simulate different loading conditions on our models, ultimately yielding biomechanical parameters. In this study, we assessed variables such as IDP values to evaluate intervertebral pressure, ROM values to assess lumbar spine mobility, facet joint stresses to evaluate the risk of facet joint degeneration, and posterior structure stresses to assess the risk of lumbar laminae fractures. These variables are closely associated with lumbar spine degenerative changes and the occurrence of postoperative complications. It is on the basis of changes in these parameters that we assess clinical improvement.
The discussion should be briefly integrated giving your results some clinical lapels. To do that, I suggest the following references
A: We appreciate your suggestion to integrate our results with clinical aspects in the discussion section. Your recommended references are pertinent, and we have incorporated them into the revised manuscript (ref. 26 & ref. 27).
Thank you once again for your constructive feedback. We are dedicated to improving our paper based on your recommendations, and we will make the necessary revisions to enhance the overall quality and clarity of our research. We are sincerely grateful for your review and your dedication to advancing the standards of scholarly research.

Reviewer 3 Report
Intro: There is no urgent requirement for a nonfusion technique. Pls modify the sentence
The purpose of study otherwise well elaborated. Must be shortened.
Methods: Well explained. Illustrations are clear.
There should have been a model depicting fusion too, as it is still the gold standard
Results: Pls correct spelling errors
The results section needs to be elaborated under subheadings, as the images alone are hard to follow
Discussion: Pls clarify the study purpose clearly
Very long. May be presented under specific subheadings
Pls elaborate the relevant literature comparing fusion vs nonfusion modalities. Also elaborate on other nonfusion modalities
Limitations need to be clearly explained; and must include that fusion was not included as a control model
Conclusion: Must be precise and shorter. Only direct observations of the study must be presented
Overall well written
Author Response
We would like to extend our sincere gratitude for taking the time to review our manuscript. Your feedback and insights are invaluable in enhancing the quality and comprehensiveness of our research. We appreciate your thoughtful suggestion and I will respond to your suggestions one by one.
Intro: There is no urgent requirement for a nonfusion technique. Pls modify the sentence
A: We would like to extend our sincere appreciation for your diligent review of our manuscript. Upon careful consideration of your feedback, we have indeed recognized the need for refinement. Therefore, we have revisited the original text in question to “Hence, there is a need for a new device that can effectively slow down the progression of lumbar degenerative disease while minimizing complications, particularly in patients with osteoporosis.”, the details are in lines 45-47.
The purpose of study otherwise well elaborated. Must be shortened.
A: We shortened the description of the purpose of the study in the introduction. The new changes are in lines 92-94 and lines 295-296 of the manuscript.
Methods: There should have been a model depicting fusion too, as it is still the gold standard
A: Thank you very much for your suggestion that fusion as a gold standard is discussed in various studies. However, we did not carry out experiments related to fusion due to the fact that we were discussing the feasibility of non-fusion techniques. Therefore, in the manuscript we used the IPS (Cofelex-F), which is also a posterior spinal column fixation, as a control group instead of fusion. Based on your suggestion, we added a comparison of fusion and nonfusion modalities in the discussion section through literature research (lines 316-326). Also, we stated in the limitations that the study did not further discuss the limitations of the fusion model as a control trial (lines 418-419).
Results: Pls correct spelling errors
The results section needs to be elaborated under subheadings, as the images alone are hard to follow
A: We have made significant changes to the results section and added subheadings to detail them. We have rechecked our spelling and have fixed the original spelling mistakes, thank you very much for your suggestions.
Discussion: Very long. May be presented under specific subheadings
A: We have revised the discussion section and its backward direction to make the discussion more comprehensive and clearly structured. The revised structure is as follows: the first paragraph of the discussion section summarizes the results of the study. Paragraph 2-3 discusses the common treatment methods for degenerative lumbar spine diseases. Paragraph 4 discusses screw insertion points. Paragraph 5 discusses unilateral and bilateral fixation. Paragraph 6 discusses the ISES technique and the Coflex rivet technique. Paragraph 7 discusses the application of the ISES technique. Paragraph 8 discusses the limitations of the study. Based on the above structure, we preferred not to add additional subheadings as this is not common in discussion.
Pls elaborate the relevant literature comparing fusion vs nonfusion modalities. Also elaborate on other nonfusion modalities
A: The introduction of fusion and nonfusion, and other nonfusion techniques was supplemented in lines 305-334 of the Discussion section.
Limitations need to be clearly explained; and must include that fusion was not included as a control model
A: The limitation "lack of integrated models as a control" has been added to the limitations (line 418-419).
Conclusion: Must be precise and shorter. Only direct observations of the study must be presented
A: Thank you very much for your recommendation. We have modified the Conclusion to make them more concise (line 442-445).
We would like to express our sincere gratitude for your valuable feedback and your diligent evaluation of our manuscript. Your insights have been pivotal in enhancing the quality and reproducibility of our study. Once again, We are sincerely grateful for your review and your dedication to advancing the standards of scholarly research.

Reviewer 4 Report
The authors have developed a well-conducted and well-written study to investigate the feasibility of a novel interlaminar screw elastic spacer technique for the treatment of lumbar degenerative disease from a biomechanical perspective.
However, I would like to make some observations before recommending your work for publication.
1. Could the authors improve the visibility of figure 2?
2. Please add updated information on how demographic and specific factors may influence the prevalence of low back pain and low back degenerative disease.
3. When you comment in the Introduction /Discussion section on osteoporosis, it would be appropriate to mention conservative treatment. There is a recent systematic review detailing the effects of therapeutic exercise on the variables that measure osteoporosis that I recommend you comment on: DOI: 10.3390/jcm10112229
4. Could the authors provide a graphical abstract of the study?
5. To enrich the Introduction/Discussion section, I recommend that the authors comment on the influence of expectations on the results of the treatment of patients with low back pain: doi.org/10.47197/retos.v46.93950
6. To enrich the Introduction/Discussion section, compare your study with other similar studies in terms of the most relevant findings.
7. Could you add a section on "Clinical Implications"?
Dear Editor,
Thank you very much for trusting me for this important task.
The authors have developed a well-conducted and well-written study to investigate the feasibility of a novel interlaminar screw elastic spacer technique for the treatment of lumbar degenerative disease from a biomechanical perspective.
I strongly believe that this work contributes to the clinical knowledge, understanding and decision making of patients, as well as highlighting the necessary research that needs to be done for a better understanding and utilization of the therapeutic approach for patients with lumbar degenerative diseases.
I recommend its publication after the authors answer me or apply some suggestions that I request them.
Kind regards,
Author Response
We would like to extend our sincere gratitude for taking the time to review our manuscript. Your feedback and insights are invaluable in enhancing the quality and comprehensiveness of our research. We appreciate your thoughtful suggestion and I will respond to your suggestions one by one.
Could the authors improve the visibility of figure 2?
A: In order to facilitate better observation of the differences between the experimental and control groups, we have divided the original Figure 2 into three parts (Figure 2, Figure 3, Figure 4) and added explanatory text next to the figures to assist readers in better understanding their content.
Please add updated information on how demographic and specific factors may influence the prevalence of low back pain and low back degenerative disease.
A: Based on your suggestion, we have incorporated additional information in lines 31-34 of the manuscript concerning how demographic and specific factors can impact the prevalence of low back pain and degenerative disease.
When you comment in the Introduction /Discussion section on osteoporosis, it would be appropriate to mention conservative treatment. There is a recent systematic review detailing the effects of therapeutic exercise on the variables that measure osteoporosis that I recommend you comment on: DOI: 10.3390/jcm10112229
A: At your suggestion, in the Discussion section (lines 409-411), we incorporated literature references and suggested in the manuscript that exercise therapy for osteoporosis be used in combination as an adjunct to spine surgery.
Could the authors provide a graphical abstract of the study?
A: I am very sorry that due to our current resource and capacity constraints we are unable to create a satisfactory graphical abstract for this paper. This is not mandatory for this journal and if we could, we would prefer not to provide a graphical abstract this time around as we do not have the experience to draw it. While we are unable to provide a graphical abstract, we provided a concise textual abstract that highlights the key points, findings, and significance of the research. This will offer readers a clear understanding of the study.
To enrich the Introduction/Discussion section, I recommend that the authors comment on the influence of expectations on the results of the treatment of patients with low back pain: doi.org/10.47197/retos.v46.93950
A: Your suggestions have enriched our discussion, and we have added relevant content to lines 309-311 of the manuscript, citing the references you provided. Thank you again for your recommendation.
To enrich the Introduction/Discussion section, compare your study with other similar studies in terms of the most relevant findings.
A: We have made major revisions and additions to both the introduction and discussion sections. Among other things, we compared the methodology and results of this study with previous studies and proved that our results have a similar trend to the results of previous studies. However, since our technique is a new one and there are no studies of the same, a direct comparison cannot be made.
Could you add a section on "Clinical Implications"?
A: We have enhanced the description of the clinical implications of the study in the introduction, discussion and conclusion. The drawbacks of fusion techniques have led to the rise of non-fusion techniques. However the current non-fusion techniques are not mature. This study is to demonstrate the feasibility of a new non-fusion fixation modality, which can be of great clinical value by providing recommendations for surgical improvement and implant design. We have interspersed references to the above clinical implications throughout the manuscript, and therefore we do not feel the need to specifically include a section on the clinical implications.

Round 2
Reviewer 1 Report
Most of the questions asked were answered by the authors. This reviewer congratulates the authors of this article for the quality of the research presented.
No comments
Reviewer 2 Report
Dear Authors,
thank you for the efforts to follow my suggestions.
No further corrections are needed.
Best regards
Reviewer 4 Report
I agree with the new version of the author’s manuscript.
Congratulations
No comments